# Representational Alignment between Deep Neural Networks and Human Brain in Speech Processing under Audiovisual Noise

## Abstract

Speech recognition in the human brain is an incremental process that begins with acoustic processing and advances to linguistic processing. While recent studies have revealed that the hierarchy of deep neural networks (DNNs) correlates with the ascending auditory pathway, the exact nature of this DNN-brain alignment remains underexplored. In this study, we investigate how DNN representations align with the brain's acoustic-to-linguistic processing. Specifically, we employed neural encoding models to simulate neural responses to acoustic (i.e., speech and noise envelope) and linguistic features (i.e., word onset and surprisal). By applying representational similarity analysis (RSA), we quantified the similarity between these neural responses and the DNN embeddings generated by a pre-trained automatic speech recognition (ASR) model, both before and after fine-tuning on audiovisual noisy data. Our results demonstrate significant DNN-brain alignment: embeddings from shallow layers exhibit higher similarity to neural responses associated with acoustic features, while those from deeper layers align more closely with neural responses related to linguistic features. Importantly, the audio-visual fine-tuning process enhances this alignment by improving noise processing in shallower layers and refining linguistic representations in deeper layers. These results suggest that fine-tuned DNN models can naturally develop human-like processing patterns in noisy environments, highlighting a functional alignment between the human brain and DNNs in speech representation.

## 1 Introduction

Speech comprehension involves multiple levels of analyses, including the encoding of basic sensory features and the extraction of more abstract linguistic information (Hickok & Poeppel, 2007). Recently, transformer-based deep neural network (DNN) models have shown human-level performance in automatic speech recognition (ASR) tasks (Baevski et al., 2020; Hsu et al., 2021; Radford et al., 2023). However, it remains unclear whether this comparable performance reflects similar internal information representations. According to a recent review, the similarity between the human brain and DNN models can be evaluated through: (1) the responses of human and DNN models to the same stimuli, (2) the stimulus features evoking the similar responses of human and DNN models, and (3) the computations involved in processing the same stimulus to yield similar response (Schyns et al., 2022). To explore this representational similarity, we correlate the representations in DNN hidden layers with the neural responses to acoustic and linguistic features, aiming to explore the similarity between DNN models and the human brain in terms of computational equivalence.

Accumulating studies provide evidence for the representational alignment between DNN models and the human brain in speech recognition. Recent studies found that the embeddings from the middle layers in DNN models best predict neural activity in the primary cortex, while those from deep layers best predict activity in the non-primary cortex (Kell et al., 2018; Li et al., 2023; Millet et al., 2022; Tuckute et al., 2023; Vaidya et al., 2022). Furthermore, the embeddings of the shallower layers are primarily predicted by acoustic features, whereas embeddings of deeper layers onward are better predicted by phonetic features (Li et al., 2023). Given that the ascending auditory pathway correlates with acoustic-to-linguistic processing in speech comprehension (Hickok & Poeppel, 2007; Giraud & Poeppel, 2012), the correlation between the DNN hierarchy and ascending auditory

pathway suggests a representational alignment between DNN hierarchy and acoustic-to-linguistic processing in the human brain. However, previous research has primarily focused on the relationship between DNNs and the auditory pathway without distinguishing between acoustic and linguistic processing in either the brain or DNNs, leaving direct evidence for this representational alignment still lacking. Moreover, most existing research has examined DNN-brain alignment exclusively in fine-tuned models optimized for specific tasks, which leaves the impact of the fine-tuning process on DNN-brain alignment unclear.

Additionally, speech is often accompanied by audiovisual noise in real-world scenes. For instance, when speaking on the phone on a busy street, we not only hear speech from the phone but also background noise from the surroundings, with visual information often unrelated to the speech. Previous studies have typically investigated the representational similarity between the human brain and DNN models using audio-only speech recognition tasks (Kell et al., 2018; Li et al., 2023). Little is explored about the DNN-brain representational similarity in audiovisual scenes. A recent study has proposed a transformer-based audiovisual automatic speech recognition (AV-ASR) framework, called AV-HuBERT (Shi et al., 2022a). The framework offers a stable training procedure to fuse audio and visual representations, and achieves promising performances on audiovisual scenes, where the visual information is related to the target speaker (e.g., lip movements) (Shi et al., 2022b) or the background noise (e.g., interfering objects) (Luo et al., 2024). It provides a useful tool for exploring the representational similarity between the human brain and DNN models in speech recognition under a more natural condition.

This study aims to investigate the alignment between representations the human brain and DNN models in speech recognition tasks under audiovisual noise. Specifically, we applied a comparative analysis of neural responses and DDN embeddings for English speech with audiovisual noise in natural settings. For DNN models, we extracted the embedding vectors from hidden layers for both pre-trained and fine-tuned models in response to the audio-only or audiovisual inputs. For neural responses, we recorded electroencephalography (EEG) signals while participants performed a speech comprehension task under audiovisual noise. To evaluate the representational similarity between model embeddings of hidden layers and the neural representations of multi-level speech processing, we quantified the neural responses to acoustic (i.e., speech envelope and noise envelope) and linguistic (i.e., word onset and word surprisal) features using neural encoding models. We then separately simulated the neural responses to these features and computed the representational similarity between the simulated neural responses and DNN embeddings (Fig.1). The main contributions of this work include:

- This work introduces a novel method that combines neural encoding model and representational similarity analysis to evaluate the representational similarity between DNN and neural representations in speech recognition at specific processing levels (e.g., acoustic and linguistic levels).
- This work reveals representational alignments between the transformer-based DNN and human brain in speech recognition under audiovisual noise: DNN representations in the shallower hidden layers are more similar to neural representations of acoustic features, whereas those in deeper layers are more similar to neural representations of linguistic features.
- This work demonstrates that the fine-tuning process enhances the DNN-brain alignments for acoustic-to-linguistic processing in noisy environments. Specifically, the fine-tuning process modulates the trends of representational similarity for acoustic and linguistic features across layers, increasing the representation of noise processing in shallower layers while refining linguistic representations in deeper layers.

## 2 APPROACH

### 2.1 SPEECH STIMULI

Stimuli consisted of 16 English speech audios selected from LRS3 corpus (Afouras et al., 2018). Each speech audio was approximately one minute long (58.57 ± 2.44 seconds). Audiovisual noise stimuli were chosen from AVNS corpus (Luo et al., 2024). Eight audiovisual noise clips were selected from *Road* class (recorded in the road side), and the other eight were selected from *Playground* class (recorded in badminton courts, basketball courts, and athletics fields). All audiovisual noise clips were recorded at a fixed point. The video depicted natural auditory scenes, and

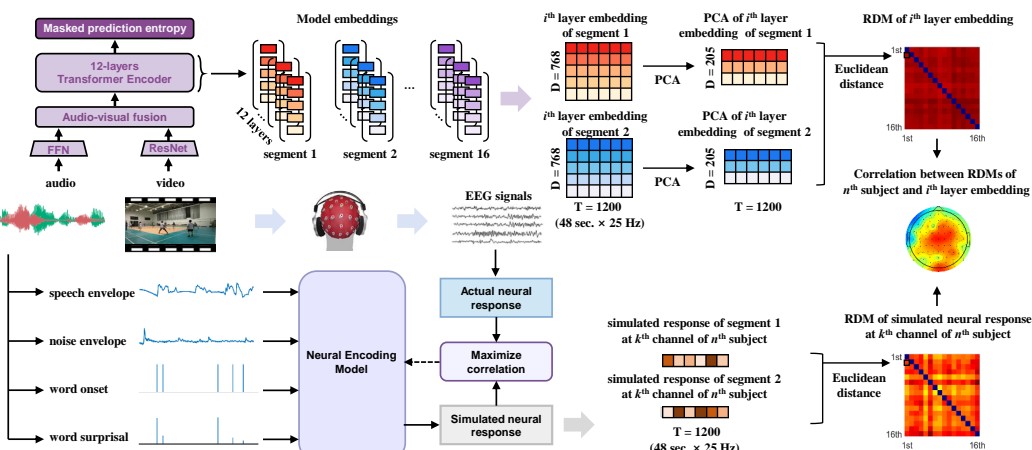

Figure 1: Schematic of analysis framework for computing representational similarity between DNNs embeddings and neural responses. For transformer-based DNNs (top panel), the model is fine-tuned on speech data with congruent audiovisual noise, starting from a pre-trained initialization. Then, embedding vectors of stimulus input are extracted from 12 transformer hidden layers, and utilized to calculate representational dissimilarity matrixs (RDMs). For neural encoding model (bottom panel), the EEG data in each EEG channel is predicted based on a linear combination of time-shifted stimulus features (speech envelope, noise envelope, word onset, and word surprisal). The encoder weights (temporal response functions) are estimated by maximizing the correlation between the actual neural responses and simulated neural responses. The simulated neural responses to stimulus features are utilized to calculate RDMs. Representational similarity is indexed through Spearman correlation between the RDMs of model embeddings and simulated neural responses.

the audio contained sounds from objects in the scenes (such as car driving and telephone ringing) and human vocalizations (such as yelling and cheering) but no speech. The noise-free speech audio recordings were overlapped with the noise audio recordings at a 0 dB signal-to-noise ratio (SNR). The noise speech audio was paired with videos either congruent or incongruent with the noise audio, resulting in congruent and incongruent audiovisual conditions. The audio recordings were sampled at 44.1 kHz, and the video recordings had a frame rate of 25 Hz. Each noise video was paired with congruent and incongruent noise audio once, resulting 32 speech-in-noise stimuli for the experiment.

## 2.2 EEG EXPERIMENT

### 2.2.1 PROCEDURE AND TASK

Twenty English native speakers (eight males; age: 22.4 ± 3.87 years old) were recruited for the EEG experiment. All participants self-reported normal hearing and vision, with no mental or neurological disorders. Written informed consent was obtained from all participants. The study was approved by the anonymous Institutional Review Board (Approval Number: $anonymous\ number$).

In the EEG experiment, the participants listened to speech clips with real-world noise while simultaneously viewing the noise-related background videos. Stimulus presentation was controlled using Psychtoolbox in Matlab software (version R2021a) (Brainard & Vision, 1997). The audio stimuli were presented binaurally via inserted earphones at a comfortable volume. The video stimuli were displayed on a monitor with a resolution of 1920×1080. After each stimulus, they answered a single-choice question with two options about the speech content by pressing buttons. Following the single-choice question, an additional oral report task was assigned for four randomly selected stimuli. This task required participants to orally describe the background videos, which ensured their attention to the visual stimulus. Participants took a 30-second break between stimuli, each of which was presented once in a random order. As a result, the accuracy rate for the single-choice task was 89.85 ± 7.74%.

### 2.2.2 EEG ACQUISITION AND PREPROCESSING

EEG signals were recorded using a 64-channel BioSemi ActiveTwo EEG system at a sample rate of 2048 Hz. Scalp electrodes were placed according to the 10-10 system, with two external electrodes placed on bilateral mastoids for re-referencing. Four external electrodes were placed on the bilateral

outer corners, and above and below the right eye to record the electrooculogram (EOG). Both EEG and EOG recordings were resampled to 128 Hz and bandpass filtered between 0.5 and 30 Hz using a linear-phase finite impulse response (FIR) filter with an order of 768. The delay caused by the filter was compensated. Filtered EEG and EOG recordings were referenced to the average recording of the bilateral mastoids, and EOG signals were linearly regressed out from all channels. Independent component analysis (ICA) was performed using EEGLAB (Delorme & Makeig, 2004) to further remove artifacts. Data exceeding 6 times the median of amplitude were clipped. Finally, the EEG recordings were bandpass filtered between 1 to 10 Hz using a zero-phase-shift filter and resampled to 25 Hz to match the sampling rate of model embeddings.

## 2.3 DNN ARCHITECTURE AND EMBEDDINGS

### 2.3.1 MODEL ARCHITECTURE

The DNN model based on the AV-HuBERT framework was constructed for automatic speech recognition in audiovisual scenes. The AV-HuBERT was a representation learning framework (Shi et al., 2022a;b), achieving state-of-the-art performance on unimodal (Shi et al., 2022a) and multimodal (Shi et al., 2022b) setups. Here, we used AV-HuBERT BASE as the model architecture for our experiment, and the framework comprised four modules: a feed-forward network audio feature extractor, a modified ResNet video feature extractor, a fusion module, and a Transformer backend with 12 layers. The FFN learned audio representation from the mel-frequency cepstral coefficients (MFCC) of the synthetic speech audio, while the ResNet learned visual representations from the raw image of the background video. These two feature extractors generated frame-level representations for the corresponding streams that were concatenated by the fusion module to form initial audiovisual features. The concatenated audiovisual representations were then fed into transformer-encoder blocks to generate an output representation list $e_{1:T}$. The embedding dimension, feed forward dimension, and attention heads of each transformer block were 768, 3072, and 12, respectively. A linear classifier was used to predict tokens with respect to each $e_i$.

As a control setup for the audio-visual model, we employed an audio-only version by masking the video input and retaining only the audio input in the current framework (Hsu et al., 2021).

### 2.3.2 PRE-TRAINED AND FINE-TUNED MODELS

For the pre-trained model, we directly utilized the open-source AV-HuBERT BASE model (via github) from Shi et al. (2022a;b) in our experiment setup. This model was pre-trained on 1326 hours of VoxCeleb2 and 433 hours of LRS3 datasets, capturing effective audiovisual speech representation without additional noise training. For the fine-tuned model, we adopted a noise-related audiovisual dataset to fine-tune the model from the pre-trained initialization (Shi et al., 2022a). Following the procedure of a previous study (Luo et al., 2024), the audiovisual dataset was created based on AVNS, and LRS3 datasets, which served as the source of background sound and of target speech respectively. Specifically, the original speech audio was extracted from LRS3 and superimposed by the background sound from the AVNS dataset with 0 dB SNR. Then, video congruent with background sound was attached to the synthetic speech audio. In total, a 433-hour synthetic dataset was generated, and the training, validation, and test sets contained 75%, 15%, and 10% sample clips, respectively. Based on this dataset, we conducted two versions of fine-tuning process: one for the audio-only model using only audio inputs, and another for the audio-visual model using both audio and visual inputs.

The audio and video inputs were downsampled to feature sequences at a frame rate of 25 Hz. The attention-based sequence-to-sequence cross-entropy loss (S2S) was used to fine-tune the model. Specifically, a transformer-based decoder was appended to the output of the encoder and decoded the $e_{1:T}$ to a sequence of target token probabilities in an autoregressive manner. The model was trained by minimizing the cross-entropy loss $L_{s2s} = \sum_t^s p(w_t|w_{1:t-1}, e_{1:T})$, where $w_t$ was the true token. Through the fine-tuning process, the speech recognition accuracy under background noise reached 53.3% for the audio-only model and 77.1% for the audio-visual model. Both models maintained reasonable performance on clean speech, with an accuracy of 91.3% and 87.3%, respectively.

### 2.3.3 MODEL EMBEDDINGS

The audio and video stimuli employed in the EEG experiment were processed by the pre-trained and fine-tuned models. The embedding vector, i.e., a 768-dimensional column vector, for each frame was extracted from each transformer layer. Moreover, the principal component analysis (PCA) was applied to mitigate the high dimensionality and redundancy in embedding vectors. For each clip and transformer layer, the embedding dimension was reduced from 768 to 205, which captured over 90% explained variance. Consequently, the embedding vectors of 16 clips were obtained from audiovisual congruent and incongruent conditions respectively.

## 2.4 NEURAL ENCODING MODEL

### 2.4.1 FEATURE EXTRACTION

Acoustic features consisted of speech envelope and noise envelope. The broadband amplitude envelope of both speech and noise waveforms were obtained through full-wave rectification. The sound envelope was then bandpass filtered between 1 to10 Hz and resampled to 25 Hz to match the sampling rate of model embeddings. Linguistic features consisted of word onset and word surprisal. Word onset was extracted using the Montreal Forced Aligner (McAuliffe et al., 2017), and represented as one-dimensional sequence with impulses aligned to the time bins of word onsets (Fig.1). Word surprisal, calculated as the negative logarithm of the conditional probability of the considered word given all the preceding words, was extracted using GPT-2 (Radford et al., 2019). Word surprisal was represented as a one-dimensional sequence with impulses at time bins of word onsets, weighted by the corresponding word surprisal values (Fig.1).

### 2.4.2 TRF ESTIMATION

A time-delayed neural encoding model, known as multivariate temporal response function (mTRF), was employed to construct the mappings between speech feature sequences and EEG responses (Ding & Simon, 2012; 2013; Crosse et al., 2021). The feature sequences and EEG responses were temporally aligned for training the neural encoding model. For each speech stimulus, the initial and last 1-second segment of feature sequences and EEG responses was excluded to avoid the onset/offset effects. The neural encoding model was fitted as $r(t) = \sum_k \sum_\tau s_k(t - \tau) \cdot w_k(\tau) + \varepsilon(t)$ for each channel and participant, where $r(t)$ was the EEG response from the channel, $s_k(t - \tau)$ was the speech feature sequence $k$ at time $t - \tau$, $w_k(\tau)$ was the regression weight for feature sequence $k$ at time lag $\tau$, and $\varepsilon(t)$ was the noise. The time lags ranged from -200 ms to 800 ms. The regression weight $w_k(\tau)$ was computed using the least squares estimation with $L2$ regularization. The regularization parameter was tuned to provide the highest simulated accuracy, which was measured by the Pearson's correlation between the simulated neural responses and the actual responses. The procedure was evaluated using 10-fold cross validation to prevent overfitting. The optimal regularization parameter $\lambda$ was set to 0.01, which yielded the highest simulated accuracy averaged over all channels, and participants.

### 2.4.3 SIMULATED RESPONSES TO SPECIFIC SPEECH FEATURES

The simulated neural responses to specific speech features could be created through the neural encoding model. Specifically, the neural response, $\hat{r}(t)$ at time $t$ was computed as the convolution of the speech feature sequence $s_k(t)$ with the corresponding regression weight $w_k(\tau)$: $\hat{r}(t) = \sum_\tau s_k(t - \tau) \cdot w_k(\tau)$. The time lags ranged from 0 ms to 800 ms. In the simulation procedure, the feature sequences of speech envelope and noise envelope were normalized using z-scores. Finally, the simulated responses to speech envelope, noise envelope, word onsets, and word surprisal was obtained respectively.

## 2.5 REPRESENTATIONAL SIMILARITY ANALYSIS

Representational dissimilarity matrix (RDM) was constructed separately for neural responses and DNN embeddings. Due to the varying length of the speech stimuli (58.57 ± 2.44 seconds), the middle 48 seconds of data was retained by excluding initial and last $(duration - 48)/2$ seconds of the data segments. This procedure also helped to avoid the onset/offset neural responses. Then, the

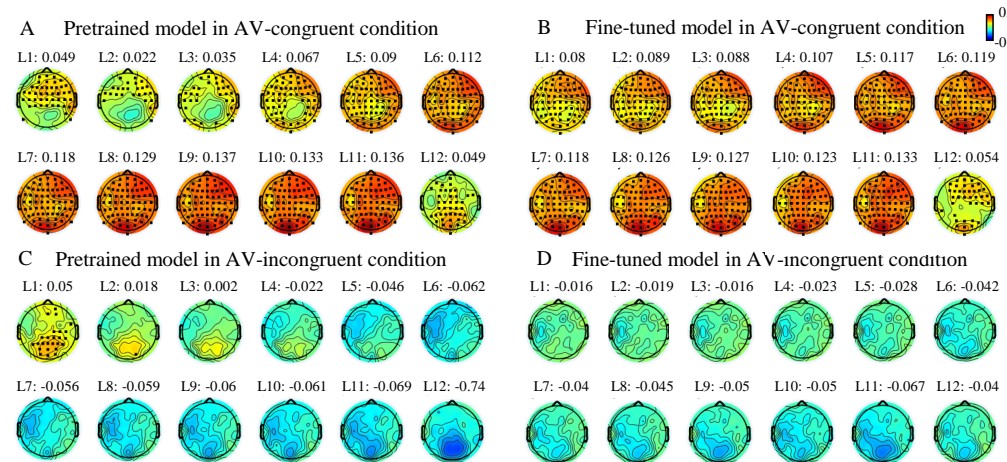

Figure 2: Topographic maps of representational similarity between DNN embeddings and actual EEG responses in speech processing with audiovisual-congruent noise (A-B) and audiovisual-incongruent noise (C-D). Black dots denote the significant channels (see Appendix for details on the statistical analysis).

Euclidean distance was calculated between the DNN embeddings or neural responses of stimulus pairs in audiovisual congruent and incongruent conditions respectively, resulting in 16×16 RDMs for each condition. For neural responses, RDMs were calculated for each participant and channel. For model embeddings, RDMs were calculated for each transformer layer. The Spearman correlation was calculated using the values in the upper triangle of the RDMs between neural responses and model embeddings. In total, 15360 correlation values (12 layers × 20 participants × 64 channels) of each condition were obtained to index the representational similarity between neural responses and model embeddings (Fig.2-4).

## 3    RESULTS

### 3.1    REPRESENTATIONAL SIMILARITY

In speech processing with audiovisual-congruent noise, significant representational similarity was observed in most of EEG channels and across all transformer hidden layers in both pre-trained (Fig.2A) and fine-tuned (Fig.2B) models. In contrast, with audiovisual-incongruent noise, significant representational similarity was limited to the first layer in pre-trained model (Fig.2C), and was absent in the fine-tuned model (Fig.2D). Given that the DNN model was fine-tuned on speech with audiovisual-congruent noise, these results suggest that DNNs can naturally evolve human-like information representations through the task-related fine-tuning process. Additionally, in speech processing with audiovisual congruent noise, the representational similarity increased from the first layer to the eleventh layer, but decreased in the last layer, which was trained to match the auto-regressive decoder for text transcriptions. Therefore, the model embeddings of the last layer (Layer 12) were excluded in the following analysis.

### 3.2    NEURAL ENCODING MODEL OF SPEECH FEATURES

On the top of representational similarity between DNN embeddings and neural responses, we explored the representational similarity between DNN embeddings and simulated neural responses to acoustic and linguistic features. These simulated neural responses were generated using neural encoding models (mTRF). The evaluation analysis revealed that all neural encoding models for acoustic and linguistic features achieved statistical significance (Fig.S1). For acoustic features, the neural encoding model of speech envelope showed a significant negative cluster at 100 ms ($T_{sum}$ = -290.92, $p$ = 0.0351) and a significant positive cluster at 200 ms ($T_{sum}$ = 340.84, $p$ = 0.0283) in fronto-central channels (Fig.S1A), which is analogous to classical N1-P2 complex (Näätänen & Picton, 1987). The neural encoding model of noise envelope showed a significant negative cluster at 400 ms ($T_{sum}$ = -1378, $p$ <0.001) in frontal channels (Fig.S1B). For linguistic features, the neural

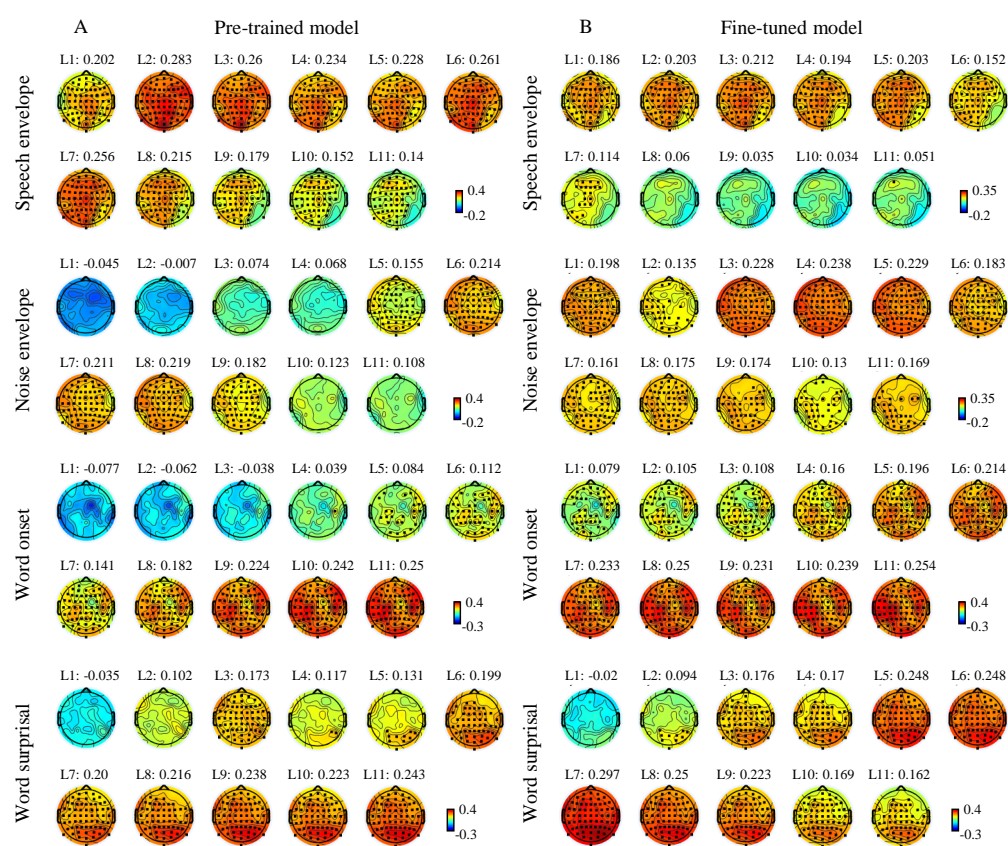

Figure 3: Topographic maps of representational similarity between DNN embeddings and simulated neural responses to speech envelope, noise envelope, word onset, and word surprisal in speech processing with audiovisual-congruent noise. Results for pre-trained model (A), and fine-tuned model (B). Black dots denote the significant channels (see Appendix for details on the statistical analysis).

encoding model of word onset showed a significant positive cluster at 200 ms ($T_{sum}$ = 929.42, $p$ = 0.0029) in fronto-central channels (Fig.S1C). The neural encoding model of word surprisal showed a significant positive cluster at 300 ms ($T_{sum}$ = 1284.6, $p$ <0.001) and 700 ms ($T_{sum}$ = 325.24, $p$ = 0.0449) in fronto-central channels and a significant negative cluster at 500 ms ($T_{sum}$ = -302.93, $p$ = 0.0488) in central channels (Fig.S1D), which is analogous to N400, the classical semantic-related ERP component (Kutas & Federmeier, 2011). In addition, we performed representational geometry validation by comparing representational dissimilarity matrices (RDMs) between actual and simulated neural responses. For each speech feature, we computed RDMs from the corresponding simulated neural responses and the actual neural responses, and then calculated the correlation between these RDMs. The significance of RDM correlation was evaluated across participants and EEG channels (Table S4). The results demonstrated that the simulated responses could preserve the representational geometry of the actual neural data. Together, all these results proved the validity of the neural encoding models of acoustic and linguistic features. Hence, these neural encoding models were employed to simulate the neural responses to acoustic and linguistic features for further representational similarity analysis.

## 3.3 FEATURE-BASED REPRESENTATIONAL SIMILARITY

To evaluate the feature-based DNN-brain alignment in speech processing, we analyzed the representational similarity between DNN embeddings and simulated neural responses to acoustic and linguistic features separately (Fig.3). Regarding acoustic features, higher representational similarity between DNN embeddings and simulated neural responses was observed in the shallower layers. For the speech envelope, the pre-trained model showed prominent representational similarity in the first eight layers, with a decrease in deeper layers (Fig.3A). The fine-tuned model exhibited higher similarity in the first six layers, with a decline observed in deeper layers (Fig.3B). For the noise enve-

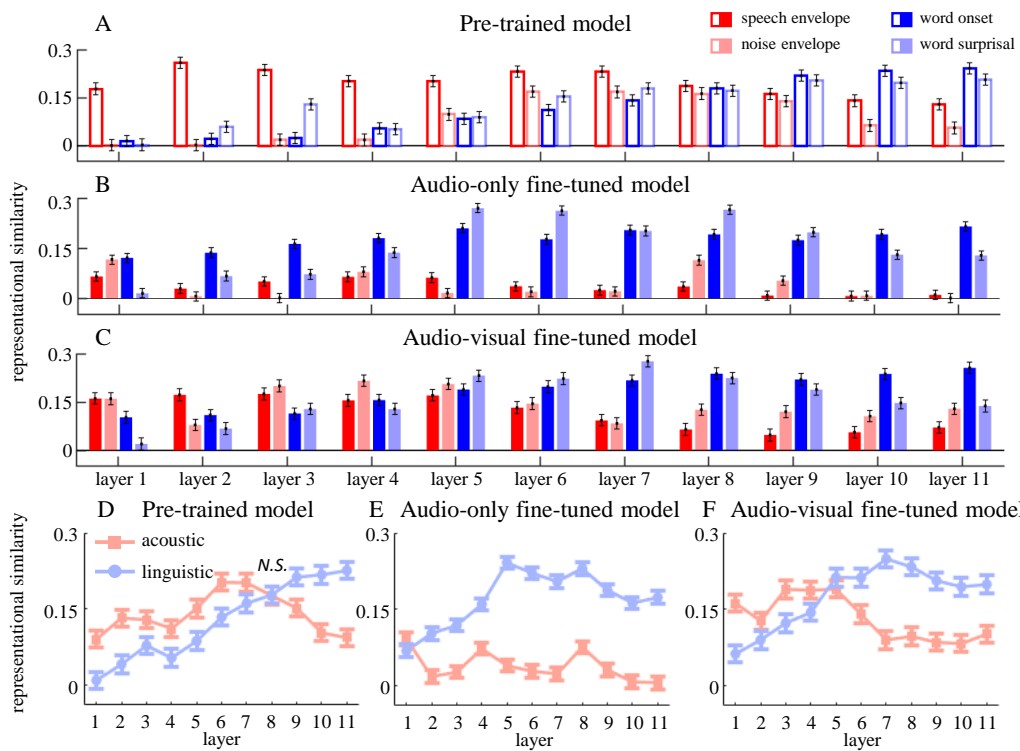

Figure 4: (A-C) Representational similarity of speech envelope, noise envelope, word onset, and word surprisal for the pre-trained, fine-tuned models, as well as their differences. The representational similarity was averaged across channels and participants. (D-E) Representational similarity for acoustic and linguistic features in the pre-trained, and fine-tuned models. The representational similarity was averaged across channels, participants and features. Error bars denote the 95% confidence interval (CI). $N.S.$ denotes non-significant.

lope, the pre-trained model demonstrated relatively higher representational similarity from Layers 5 to 9, with a peak at Layer 6 (Fig.3A). In contrast, the fine-tuned model exhibited higher similarity in the first five layers, with a peak at Layer 4 (Fig.3B). Regarding linguistic features, higher representational similarity between DNN embeddings and simulated neural responses was observed in the deeper layers. For word onsets, both the pre-trained and fine-tuned models showed increasing representational similarity from shallow to deep layers, with a peak at Layer 11 (Fig.3A&B). Similarly, for word surprisal, representational similarity increased from shallow to deep layers, followed by relative stability in the deeper layers (Fig.3A&B). These results highlight a significant DNN-brain alignment: embeddings from shallow layers exhibit higher similarity to neural responses associated with acoustic features, while those from deeper layers align more closely with neural responses related to linguistic features.

## 3.4 REPRESENTATIONAL MODULATION OF FINE-TUNING

In this section, we additionally included a representational analysis of an audio-only model as a control condition to isolate the effects of visual input during fine-tuning process (see Section 2.3). The representational similarity for the pre-trained, audio-only fine-tuned, and audio-visual fine-tuned models is summarized in Fig.4A-C. The LME analysis was applied to investigate the significant main effects of model types, layers, speech features, and their interactions (Table S1). Additionally, the simple effect analysis was applied to compare the across-layer representational similarity between the pre-trained and fine-tuned models (Table S2). Regarding acoustic features, the fine-tuning process decreased the representational similarity for speech envelope across nearly all layers (Layers 2-11) for both the audio-only and audio-visual models. Despite these reductions, the fine-tuning process did not alter the overall trend of higher representational similarity in shallower layers. Importantly, the fine-tuning process shifted the trend of representational similarity for noise envelope across layers, increasing it in shallower layers (Layers 1 and 4 for the audio-only model, and layers 1-5 for the audio-visual model) while decreasing it in deeper layers (Layers 5-11 for the audio-only

model, and layers 6-9 for the audio-visual model). This shift may indicate that fine-tuning could enhance the model's ability to disentangle noise from speech at an earlier processing stage, potentially reducing the influence of noise on higher-level processing. Regarding linguistic features, the fine-tuning process increased the representational similarity for both word onsets and word surprisal in middle layers (Layers 2-8 for both models), and preserved the trend of higher representational similarity in deeper layers (Layers 9-11 for both models).

We further compared the across-layer distribution of representational similarity between the pretrained and fine-tuned models (Fig.4 D-F; also see Table S3). Our analysis revealed that the fine-tuning process modifies the dominance of representational similarity for acoustic and linguistic features across layers, rebalancing the model's focus to improve speech processing in noisy environments. Specifically, in the pretrained model, representational similarity for acoustic features dominated from Layers 1 to 7, with a transition to linguistic features from Layers 9 to 11 (Fig.4 D). In contrast, in the audio-only fine-tuned model, this transition occurred earlier at Layer 2 (Fig.4 E), indicating a rapid progression toward higher-level linguistic representations after fine-tuning process. Notably, in the audio-visual fine-tuned model, the transition was delayed relative to the audio-only condition, occurring at Layer 5. Here, acoustic features dominated from Layers 1 to 5, and linguistic features emerged from Layers 6 to 11 (Fig.4F). This intermediate transition point indicated that visual input prolonged acoustic processing in the network, which may support robustness under noise by maintaining access to acoustic features in middle layers. Together, all these findings suggest that fine-tuned DNN models can naturally develop an acoustic-to-linguistic processing hierarchy in automatic speech recognition tasks, with multi-modal input regulating the depth of acoustic-to-linguistic transition. These results highlight a functional alignment between the human brain and DNNs in speech processing, where sensory context modulates the balance between bottom-up and top-down feature processing.

## 4 LIMITATIONS

There are two main limitations of the current work that should be noted. First, this work primarily focuses on acoustic features (e.g., speech and noise envelopes) and linguistic features (e.g., word onsets and surprisal) that are considered to exhibit universal properties across human languages. While this approach enables us to investigate fundamental mechanisms underlying DNN-brain alignment in speech processing, the exclusive inclusion of native English speakers limits the linguistic diversity of our experimental setup. It still remains crucial for future studies to incorporate a more linguistically diverse data to confirm whether these findings can be generalized across languages and populations. Second, this work presents a case study of our DNN-brain representational measuring method applied to AV-HuBERT, which is a transformer-based architecture with advanced audiovisual integration capabilities. While this choice was motivated by its relevance to our experimental goals, it raises questions regarding the extent to which the current findings depend on the specific model architecture. Future studies should extend this line of research to include comparisons across multiple DNN architectures to further validate the generalizability of our findings.

## 5 CONCLUSION

In this study, we introduced a novel method to investigate the feature-based representational similarity between DNN embeddings and neural responses in speech recognition under audiovisual noise. An important advantage of our approach lies in the use of neural encoding models (TRFs) to establish temporal mapping between stimulus features and neural responses. It enables us to generate simulated neural responses for new noise scenarios without the need to re-collect neural activity from the human brain. Therefore, our framework can be extended to evaluate DNN-brain alignment in more realistic and diverse noisy environments. The current results demonstrated that both pre-trained and fine-tuned models exhibit representational alignments with acoustic-to-linguistic processing in the human brain. Furthermore, the audio-visual fine-tuning process enhances such DNN-brain alignments by modifying the dominance of acoustic and linguistic processing across DNN layers, thus improving the model's ability to convert raw acoustic inputs into linguistic representations in noisy environments. Our findings suggest that speech representations learned in transformer-based DNNs naturally evolve toward the human-like processing patterns, providing credible evidence for the interpretability of DNNs in automatic speech recognition tasks.

## REPRODUCIBILITY STATEMENT

To facilitate the reproducibility of our work, we provide comprehensive details and resources. Key data and code are included in the supplementary materials. All publicly available datasets and code used in our experiments are explicitly cited in the main text.

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

## A APPENDIX

### A.1 STATISTICAL ANALYSIS

The significance of the neural encoding model (specifically, the mTRF model) was assessed using a cluster-based permutation test Gerber (2021) in Matlab software (version R2021a). To further investigate how representational similarity between simulated neural responses and DNN embeddings varies across layers, we employed a linear mixed effects (LME) model using $lmeTest$ package (Kuznetsova et al., 2017) in R statistical software (version 4.1.2). Additionally, simple effect analyses were conducted to evaluate differences in representational similarity across model types and speech features using $emmeans$ package (Lenth et al., 2019) in R statistical software. When multiple comparisons were performed, the $p$-value was adjusted using the false discovery rate (FDR) correction (Benjamini & Hochberg, 1995).

To evaluate the significance of neural encoding model (mTRF model), we computed surrogate mTRF models using the speech feature sequences circularly shifted a random time lag ranging from 1 to 1200 samples (1/25 to 48 seconds). This procedure was repeated for 101 times, resulting in 101 surrogate mTRF models for the actual mTRF model. All the mTRF models were baseline corrected by subtracting the averaged value at time lags from -200 to 0 ms. The actual mTRF was then evaluated by comparing it with the chance-level mTRF averaged across all the corresponding surrogate mTRF models using a cluster-based permutation test. For each time-channel sample, the actual and chance-level mTRF weights of all participants were compared using a paired $t$-test. Samples with a significant $t$-value ($p < .025$) were selected. Spatiotemporally neighboring significant samples were grouped into clusters, and the cluster-level statistic was defined as the sum of the $t$-values within each cluster. The significance of a cluster was evaluated using Monte-Carlo method: First, actual and chance-level mTRFs were collected into a single set. Second, the single set of mTRFs was randomly divided into two equal subsets. Third, the $t$-values and cluster-level statistics were calculated based on these two subsets. Fourth, the second and third steps were repeated for 1024 times, resulting in a distribution of the largest cluster-level statistics under the null hypothesis. Fifth, the actual cluster-level statistics were determined as significant if they exceeded the 95th percentile of the null distribution. As the significance of a cluster was judged against a null distribution, this procedure inherently corrected for multiple comparisons.

To evaluate the significance level of the representational similarity, surrogate correlation values were calculated by the shuffled RDMs between neural responses and model embeddings. This procedure was repeated 101 times for each participant and EEG channel. The significance level was determined as $(N + 0.01) / (101 + 0.01)$, where $N$ represents the number of times the surrogate correlation values exceeded the actual correlation values. For the condition-level analysis, the significance of representational similarity was assessed using the correlation values obtained from each participant and EEG channel. For the channel-level analysis, the significance was evaluated using the mean correlation values averaged across all participants.

To further investigate how representational similarity between simulated neural responses and DNN embeddings varies across layers, we utilized a linear mixed effect (LME) model. The model included the fixed effects of $model\ types$ (pre-trained model vs. fine-tuned model), $DNN\ layers$ (eleven levels), and $speech\ features$. The speech features were categorized into specific features (speech envelope, noise envelope, word onset, and word surprisal) or broader types (acoustic vs. linguistic). The random effects of $participants$ and $channels$ were also included in the LME model. As a result, the model was described as: $representational\ similarity \sim model\ types * DNN\ layers * speech\ features + (1\,|\,participants) + (1\,|\,channels)$. Furthermore, simple effect analyses were conducted to assess the difference in representational similarity across model types and speech features. In the LME analysis, we set insignificant representational similarity values to zero. This exclusion criterion ensured that only statistically reliable representational similarity values were included in the LME models, preserving the accuracy of our statistical analysis.

All test statistics are summarized in Table S1 to S3.

Table S1: Contrasts for the LME model with the interaction across model types, DNN layers, and speech features on the representational similarity (env.: speech envelope. n.env.: noise envelope. w.ons.: word onset. w.surp.: word surprisal). The $p$-values were adjusted using the Tukey method to control for the increased error rate associated with multiple comparisons.

**Pre-trained model**

| Contrast | Estimate | z | $p_{adjusted}$ | Estimate | z | $p_{adjusted}$ | Estimate | z | $p_{adjusted}$ | Estimate | z | $p_{adjusted}$ |
|---|---|---|---|---|---|---|---|---|---|---|---|---|
| | **Layer 1** | | | **Layer 2** | | | **Layer 3** | | | **Layer 4** | | |
| env.-n.env. | 0.178 | 31.8 | <.0001 | 0.259 | 46.2 | <.0001 | 0.220 | 39.2 | <.0001 | 0.185 | 33.1 | <.0001 |
| env.-w.ons. | 0.164 | 29.2 | <.0001 | 0.240 | 42.8 | <.0001 | 0.214 | 38.2 | <.0001 | 0.148 | 26.4 | <.0001 |
| env.-w.surp. | 0.176 | 31.4 | <.0001 | 0.202 | 36.0 | <.0001 | 0.107 | 19.1 | <.0001 | 0.152 | 27.2 | <.0001 |
| n.env.-w.ons. | -0.014 | -2.6 | 1 | -0.019 | -3.4 | 0.590 | -0.005 | -1.0 | 1 | -0.037 | -6.7 | <.0001 |
| n.env.-w.surp. | -0.002 | -0.4 | 1 | -0.057 | -10.2 | <.0001 | -0.112 | -20.0 | <.0001 | -0.033 | -5.9 | <.0001 |
| w.ons.-w.surp. | 0.012 | 2.2 | 1 | -0.039 | -6.9 | <.0001 | -0.107 | -19.1 | <.0001 | 0.004 | 0.8 | 1 |
| | **Layer 5** | | | **Layer 6** | | | **Layer 7** | | | **Layer 8** | | |
| env.-n.env. | 0.104 | 18.6 | <.0001 | 0.064 | 11.4 | <.0001 | 0.064 | 11.5 | <.0001 | 0.024 | 4.3 | .012 |
| env.-w.ons. | 0.119 | 21.2 | <.0001 | 0.121 | 21.7 | <.0001 | 0.092 | 16.4 | <.0001 | 0.007 | 1.2 | 1 |
| env.-w.surp. | 0.114 | 20.3 | <.0001 | 0.080 | 14.2 | <.0001 | 0.054 | 9.6 | <.0001 | 0.015 | 2.6 | 1 |
| n.env.-w.ons. | 0.015 | 2.7 | 1 | 0.058 | 10.3 | <.0001 | 0.027 | 4.9 | .0001 | -0.017 | -3.1 | 1 |
| n.env.-w.surp. | 0.010 | 1.7 | 1 | 0.016 | 2.8 | 1 | -0.010 | -1.9 | 1 | -0.010 | -1.7 | 1 |
| w.ons.-w.surp. | -0.005 | -0.9 | 1 | -0.042 | -7.5 | <.0001 | -0.038 | -6.7 | <.0001 | 0.008 | 1.4 | 1 |
| | **Layer 9** | | | **Layer 10** | | | **Layer 11** | | | | | |
| env.-n.env. | 0.024 | 4.3 | .017 | 0.078 | 13.9 | <.0001 | 0.074 | 13.2 | <.0001 | | | |
| env.-w.ons. | -0.057 | -10.1 | <.0001 | -0.095 | -17.0 | <.0001 | -0.113 | -20.2 | <.0001 | | | |
| env.-w.surp. | -0.042 | -7.4 | <.0001 | -0.056 | -10.0 | <.0001 | -0.078 | -13.9 | <.0001 | | | |
| n.env.-w.ons. | -0.081 | -14.4 | <.0001 | -0.173 | -30.9 | <.0001 | -0.188 | -33.5 | <.0001 | | | |
| n.env.-w.surp. | -0.065 | -11.7 | <.0001 | -0.134 | -24.0 | <.0001 | -0.152 | -27.1 | <.0001 | | | |
| w.ons.-w.surp. | 0.015 | 2.7 | 1 | 0.039 | 6.9 | <.0001 | 0.036 | 6.4 | <.0001 | | | |

**Fine-tuned model (audio-only / audio-visual)**

| Contrast | Estimate | z | $p_{adjusted}$ | Estimate | z | $p_{adjusted}$ | Estimate | z | $p_{adjusted}$ | Estimate | z | $p_{adjusted}$ |
|---|---|---|---|---|---|---|---|---|---|---|---|---|
| | **Layer 1** | | | **Layer 2** | | | **Layer 3** | | | **Layer 4** | | |
| env.-n.env. | -0.052 / 0.001 | -9.7 / 0.1 | <.0001 / 1 | 0.023 / 0.094 | 4.3 / 16.8 | .014 / <.0001 | 0.050 / -0.025 | 9.4 / -4.4 | <.0001 / .007 | -0.015 / -0.061 | -2.9 / -10.9 | 1 / <.0001 |
| env.-w.ons. | -0.056 / 0.059 | -10.6 / 10.5 | <.0001 | -0.108 / 0.063 | -20.4 / 11.3 | <.0001 | -0.113 / 0.061 | -21.4 / 10.9 | <.0001 | -0.117 / -0.001 | -22.0 / -0.2 | <.0001 / 1 |
| env.-w.surp. | 0.051 / 0.142 | 9.6 / 25.3 | <.0001 | -0.038 / 0.106 | -7.1 / 18.8 | <.0001 | -0.022 / 0.047 | -4.1 / 8.4 | <.0001 | -0.073 / 0.027 | -13.8 / 4.9 | <.0001 |
| n.env.-w.ons. | -0.005 / 0.058 | -0.9 / 10.4 | 1 / <.0001 | -0.131 / -0.031 | -24.7 / -5.5 | <.0001 | -0.163 / 0.086 | -30.8 / 15.4 | <.0001 | -0.101 / 0.060 | -19.1 / 10.7 | <.0001 |
| n.env.-w.surp. | 0.102 / 0.141 | 19.3 / 25.2 | <.0001 | -0.061 / 0.012 | -11.4 / 2.1 | <.0001 / 1 | -0.071 / 0.072 | -13.5 / 12.9 | <.0001 | -0.058 / 0.088 | -10.9 / 15.8 | <.0001 |
| w.ons.-w.surp. | 0.107 / 0.083 | 20.2 / 14.8 | <.0001 | 0.070 / 0.042 | 13.3 / 7.6 | <.0001 | 0.09 / -0.014 | 17.3 / -2.5 | <.0001 / 1 | 0.045 / 0.028 | 8.2 / 5.1 | <.0001 |
| | **Layer 5** | | | **Layer 6** | | | **Layer 7** | | | **Layer 8** | | |
| env.-n.env. | 0.046 / -0.035 | 8.8 / -6.3 | <.0001 | 0.016 / -0.013 | 3.1 / -2.2 | 1 | 0.003 / 0.009 | 0.7 / 1.6 | 1 | -0.080 / -0.062 | -15.0 / -11.0 | <.0001 |
| env.-w.ons. | -0.149 / -0.018 | -28.1 / -3.3 | <.0001 / .710 | -0.142 / -0.065 | -26.7 / -11.7 | <.0001 | -0.181 / -0.125 | -34.1 / -22.3 | <.0001 | -0.158 / -0.174 | -29.7 / -31.1 | <.0001 |
| env.-w.surp. | -0.209 / -0.062 | -39.5 / -11.1 | <.0001 | -0.228 / -0.091 | -43.0 / -16.2 | <.0001 | -0.179 / -0.185 | -33.7 / -33.0 | <.0001 | -0.232 / -0.161 | -43.7 / -28.7 | <.0001 |
| n.env.-w.ons. | -0.195 / 0.017 | -36.8 / 3.0 | <.0001 / 1 | -0.158 / -0.053 | -29.8 / -9.4 | <.0001 | -0.184 / -0.134 | -34.8 / -24.0 | <.0001 | -0.078 / -0.113 | -14.7 / -20.1 | <.0001 |
| n.env.-w.surp. | -0.256 / -0.027 | -48.2 / -4.7 | <.0001 / .002 | -0.244 / -0.078 | -46.1 / -13.9 | <.0001 | -0.182 / -0.194 | -34.3 / -34.6 | <.0001 | -0.152 / -0.099 | -28.7 / -17.7 | <.0001 |
| w.ons.-w.surp. | -0.061 / -0.043 | -11.4 / -7.8 | <.0001 | -0.086 / -0.025 | -16.2 / -4.5 | <.0001 / .005 | 0.002 / -0.060 | 0.46 / -10.6 | 1 / <.0001 | -0.074 / 0.014 | -14.0 / 2.4 | <.0001 / 1 |
| | **Layer 9** | | | **Layer 10** | | | **Layer 11** | | | | | |
| env.-n.env. | -0.045 / -0.073 | -8.4 / -13.0 | <.0001 | -0.001 / -0.050 | -0.2 / -8.9 | 1 / <.0001 | 0.010 / -0.058 | 1.8 / -10.4 | 1 / <.0001 | | | |
| env.-w.ons. | -0.167 / -0.173 | -31.5 / -30.9 | <.0001 | -0.186 / -0.182 | -35.1 / -32.5 | <.0001 | -0.206 / -0.186 | -38.9 / -33.2 | <.0001 | | | |
| env.-w.surp. | -0.190 / -0.142 | -35.9 / -25.3 | <.0001 | -0.125 / -0.091 | -23.6 / -16.3 | <.0001 | -0.119 / -0.067 | -22.5 / -12.0 | <.0001 | | | |
| n.env.-w.ons. | -0.122 / -0.101 | -23.1 / -18.0 | <.0001 | -0.185 / -0.132 | -34.9 / -23.6 | <.0001 | -0.216 / -0.128 | -40.7 / -22.8 | <.0001 | | | |
| n.env.-w.surp. | -0.145 / -0.069 | -27.5 / -12.3 | <.0001 | -0.124 / -0.041 | -23.4 / -7.4 | <.0001 | -0.129 / -0.009 | -24.3 / -1.6 | <.0001 / 1 | | | |
| w.ons.-w.surp. | -0.023 / 0.032 | -4.4 / 5.6 | <.0001 | 0.061 / 0.091 | 11.5 / 16.2 | <.0001 | 0.087 / 0.119 | 16.4 / 21.3 | <.0001 | | | |

Table S2: Simple effects analysis of representational similarity: Comparisons between pre-trained and fine-tuned models (pre-trained model vs. audio-only model / pre-trained model vs. audio-visual model) for Fig.4A-C. The $p$-values were adjusted using the Bonferroni method to control for the increased error rate associated with multiple comparisons.

| | Estimate | $z$ | $p_{adjusted}$ | | Estimate | $z$ | $p_{adjusted}$ |
|---|---|---|---|---|---|---|---|
| | | | **Speech envelope** | | | | |
| **Layer 1** | 0.113 / 0.017 | 21.4 / 3.0 | <.0001 / .128 | **Layer 7** | 0.209 / 0.140 | 39.4 / 24.9 | <.0001 |
| **Layer 2** | 0.232 / 0.087 | 43.8 / 15.6 | <.0001 | **Layer 8** | 0.153 / 0.122 | 28.8 / 21.8 | <.0001 |
| **Layer 3** | 0.187 / 0.061 | 35.3 / 11.0 | <.0001 | **Layer 9** | 0.155 / 0.115 | 29.2 / 20.5 | <.0001 |
| **Layer 4** | 0.139 / 0.047 | 26.2 / 8.4 | <.0001 | **Layer 10** | 0.135 / 0.084 | 25.5 / 15.1 | <.0001 |
| **Layer 5** | 0.141 / 0.031 | 26.5 / 5.5 | <.0001 | **Layer 11** | 0.120 / 0.058 | 22.6 / 10.4 | <.0001 |
| **Layer 6** | 0.198 / 0.100 | 37.3 / 17.9 | <.0001 | | | | |
| | | | **Noise envelope** | | | | |
| **Layer 1** | -0.116 / -0.161 | -21.9 / -28.7 | <.0001 | **Layer 7** | 0.148 / 0.085 | 27.9 / 15.1 | <.0001 |
| **Layer 2** | -0.004 / -0.077 | -0.7 / -13.8 | 1 / <.0001 | **Layer 8** | 0.049 / 0.037 | 9.2 / 6.5 | <.0001 |
| **Layer 3** | 0.018 / -0.183 | 3.3 / -32.7 | .041 / <.0001 | **Layer 9** | 0.086 / 0.018 | 16.3 / 3.3 | .0495 |
| **Layer 4** | -0.062 / -0.199 | -11.6 / -35.6 | <.0001 | **Layer 10** | 0.056 / -0.044 | 10.6 / -7.8 | <.0001 |
| **Layer 5** | 0.083 / -0.109 | 15.7 / -19.4 | <.0001 | **Layer 11** | 0.055 / -0.074 | 10.5 / -13.2 | <.0001 |
| **Layer 6** | 0.151 / 0.024 | 28.4 / 4.3 | <.0001 | | | | |
| | | | **Word onset** | | | | |
| **Layer 1** | -0.107 / -0.089 | -20.2 / -15.8 | <.0001 | **Layer 7** | -0.064 / -0.077 | -12.0 / -13.8 | <.0001 |
| **Layer 2** | -0.116 / -0.089 | -21.9 / -15.9 | <.0001 | **Layer 8** | -0.012 / -0.059 | -2.2 / -10.5 | 1 / <.0001 |
| **Layer 3** | -0.140 / -0.092 | -26.4 / -16.3 | <.0001 | **Layer 9** | 0.045 / -0.002 | 8.4 / -0.3 | <.0001 / 1 |
| **Layer 4** | -0.126 / -0.102 | -23.7 / -18.2 | <.0001 | **Layer 10** | 0.044 / -0.003 | 8.3 / -0.5 | <.0001 / 1 |
| **Layer 5** | -0.127 / -0.107 | -24.0 / -19.0 | <.0001 | **Layer 11** | 0.027 / -0.015 | 5.1 / -2.6 | <.0001 |
| **Layer 6** | -0.065 / -0.087 | -12.3 / -15.5 | <.0001 | | | | |
| | | | **Word surprisal** | | | | |
| **Layer 1** | -0.012 / -0.018 | -2.2 / -3.1 | 1 / .073 | **Layer 7** | -0.024 / -0.099 | -4.4 / -17.7 | <.0001 |
| **Layer 2** | -0.007 / -0.008 | -1.3 / -1.5 | 1 / 1 | **Layer 8** | -0.094 / -0.053 | -17.7 / -9.4 | <.0001 |
| **Layer 3** | 0.058 / 0.001 | 11.0 / 0.2 | <.0001 / 1 | **Layer 9** | 0.006 / 0.014 | 1.1 / 2.6 | 1 / .427 |
| **Layer 4** | -0.087 / -0.078 | -16.3 / -13.9 | <.0001 | **Layer 10** | 0.066 / 0.049 | 12.5 / 8.8 | <.0001 |
| **Layer 5** | -0.183 / -0.145 | -34.4 / -25.9 | <.0001 | **Layer 11** | 0.078 / 0.069 | 14.8 / 12.3 | <.0001 |
| **Layer 6** | -0.110 / -0.070 | -20.6 / -12.5 | <.0001 | | | | |

Table S3: Simple effects analysis of representational similarity: Comparisons between acoustic and linguistic features for Fig.4D-F. The $p$-values were adjusted using the Bonferroni method to control for the increased error rate associated with multiple comparisons.

| | Estimate | $z$ | $p$ adjusted | | Estimate | $z$ | $p$ adjusted |
|---|---|---|---|---|---|---|---|
| | | | **Pre-trained** | | | | |
| **Layer 1** | 0.081 | 19.6 | <.0001 | **Layer 7** | 0.041 | 9.8 | <.0001 |
| **Layer 2** | 0.091 | 22.1 | <.0001 | **Layer 8** | -0.001 | -0.3 | 1 |
| **Layer 3** | 0.051 | 12.3 | <.0001 | **Layer 9** | -0.061 | -14.8 | <.0001 |
| **Layer 4** | 0.058 | 13.9 | <.0001 | **Layer 10** | -0.115 | -27.8 | <.0001 |
| **Layer 5** | 0.064 | 15.6 | <.0001 | **Layer 11** | -0.133 | -32.1 | <.0001 |
| **Layer 6** | 0.069 | 16.6 | <.0001 | | | | |
| | | | **Fine-tuned (audio-only / audio-visual)** | | | | |
| **Layer 1** | 0.023 / 0.100 | 5.8 / 24.2 | <.0001 | **Layer 7** | -0.182 / -0.160 | -46.2 / -38.7 | <.0001 |
| **Layer 2** | -0.085 / 0.037 | -21.5 / 9.1 | <.0001 | **Layer 8** | -0.155 / -0.137 | -39.5 / -33.1 | <.0001 |
| **Layer 3** | -0.092 / 0.067 | -23.5 / 16.1 | <.0001 | **Layer 9** | -0.156 / -0.121 | -39.8 / -29.4 | <.0001 |
| **Layer 4** | -0.087 / 0.044 | -22.2 / 10.6 | <.0001 | **Layer 10** | -0.155 / -0.112 | -39.5 / -27.1 | <.0001 |
| **Layer 5** | -0.202 / -0.023 | -51.5 / -5.5 | <.0001 | **Layer 11** | -0.168 / -0.097 | -42.7 / -23.6 | <.0001 |
| **Layer 6** | -0.193 / -0.072 | -49.2 / -17.4 | <.0001 | | | | |

Table S4: Representational geometry validation between simulated and actual neural responses. The averaged number of EEG channels with significant representational similarity, the RDM correlation (Spearman's $r$), and the corresponding FDR-corrected $p$-value (permutation test) are reported.

| Speech feature | Channel count | RDM correlation | $p$ adjusted |
|---|---|---|---|
| Speech envelope | 14.4 | $0.31 \pm 0.10$ | < .048 |
| Noise envelope | 14.5 | $0.31 \pm 0.09$ | < .048 |
| Word onsets | 15.5 | $0.30 \pm 0.09$ | < .045 |
| Word surprisal | 16.5 | $0.37 \pm 0.11$ | < .006 |

## A.2 SUPPLEMENTARY FIGURES

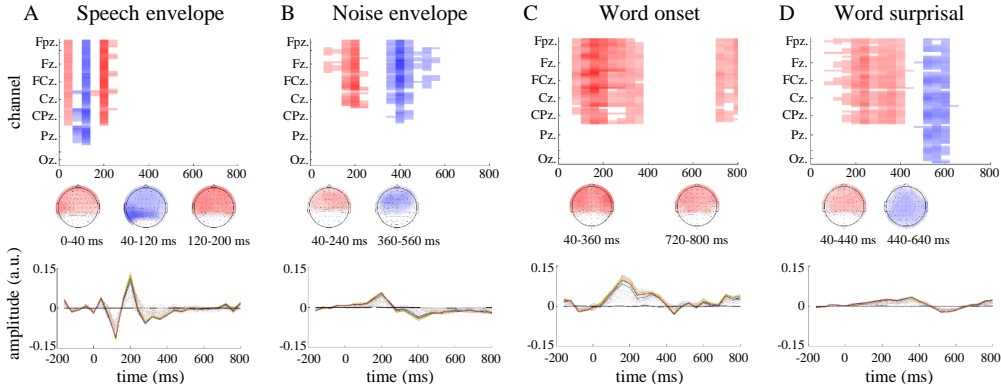

Figure S1: Neural encoding model weights (mTRFs) for speech envelope (A), noise envelope (B), word onset (C), and word surprisal (D). The top panels show the spatiotemporal cluster of mTRFs. The horizontal x-axis denotes time lags of mTRFs, and the vertical y-axis denotes 64 channels from occipital (bottom) to frontal (top). Each brick in the coordinate system represents the mTRF amplitude per time interval and channel. Significant positive and negative amplitude are displayed in red and blue, respectively. The topographies show the spatial distribution of significant spatiotemporal clusters. The topographies show the spatial distribution of significant spatiotemporal clusters. Color bar represents $t$-value obtained in the cluster-based permutation test. The bottom panels show the amplitude of mTRFs. The colorful lines represent the amplitude of mTRF in each channel. The black lines represent the amplitude of chance level (surrogate mTRF).

---

**Welcome to the Experiment!**

This study aims to explore the cognitive mechanisms of speech comprehension.

**Task Overview**

- You'll listen to 32 English speech segments, each lasting about 1 minute.
- A video will accompany the speech.
- Before each speech, an instruction will prompt you to press the SPACE button when ready.
- Your EEG signals will be recorded throughout the experiment.

**During the Experiment**

- Keep your head and body still while listening and watching (Movements can disrupt the signals).
- Pay close attention to both the speech and video when present.
- When there's no video, focus on the speech and fixate on the red "+".
- After the speech, a choice judgment will appear (press "A" or "B"). Once chosen, the judgment disappears.
- In certain rounds, you will be required to provide a verbal description of the video. After completing the description, press the SPACE button to continue.
- After a 5-second rest, the next round begins.

**Breaks**

- Enjoy a at least 30-second rest after every 16 speech segments.
- After the break, press SPACE when ready to continue.

**Compensation**

Upon completing the experiment, you will receive a compensation of 200 RMB, which will be paid via bank transfer.

Thank you for your participation! If you have any questions, feel free to ask.

---

Figure S2: The full text of instructions given to participants.

## A.3 THE USE OF LARGE LANGUAGE MODELS (LLMS)

This work does not utilize large language models (LLMs) as assistants.

