# OpenReview forum: "Representational Alignment between Deep Neural Networks and Human Brain in Speech Processing under Audiovisual Noise"
_ICLR.cc/2026/Conference — ICLR 2026 Conference Withdrawn Submission_

### Official Review · Reviewer_Q6SF · 2025-10-22

**Soundness:** 2
**Presentation:** 2
**Contribution:** 2
**Rating:** 4
**Confidence:** 5

**Summary:**

This work is part of a broader research effort to explore the alignment between representations from deep neural network models (DNNs) and human brain activity during speech comprehension. Specifically, the authors examine the hierarchy of language processing from early auditory pathways to higher-level language cognition, in both a DNN (AVHuBERT model) and human brain recordings (EEG) while participants listen to speech under naturalistic audiovisual conditions. The evaluation focuses on building neural encoding by linearly mapping interpretable four features (speech-envelope, noise-envelope, word onset and word surprisal) to EEG on a per channel, per participant basis, and compute the representational dissimilarity matrices (RDMs) between actual and predicted neural responses. Also, the authors use RDMs to assess the similarity between DNN embeddings of hidden layers and neural responses for pretrained, and fine-tuned speech recognition models.

**Contributions:**

* *Introduction to novel stimulus condition setup:* This study presents a novel setup for audiovisual congruent and incongruent conditions while participants listen to speech.
* *Introduction to new EEG dataset:* This study presents a new naturalistic EEG dataset in which participants listened to speech clips with real-world noise while simultaneously viewing the noise-related background videos.
* *Comprehensive evaluation:* The study presents an extensive assessment of both pretrained and finetuned speech models and evaluates their hierarchy of information processing. Further, the fine-tuned models are created with an audio-only model using only audio inputs, and another for the audiovisual model using both audio and visual inputs. Using these settings, the authors measure the alignment between DNN embeddings and EEG responses across layers.

**Technical summary:**
This is primarily an empirical study, and its methodology involves the following components:
* *Data collection:* The authors collected a new EEG dataset, with participants simultaneously viewing noise-related videos during speech-in-noise.
* *mTRF neural encoding model:* The four interpretable features (speech-envelope, noise-envelope, word onset and word surprisal) are used in an encoding model to predict EEG response per channel and per participant. The temporal alignment between speech stimuli and EEG responses involves delaying of signal (i.e., temporal lags from -200 ms to 800 ms), where the 48 secs speech stimulus is sampled at 25 Hz (1200 samples x  4 features x 26 [lags]) as input, while 1200 samples x 64 [EEG channels]) as the target. A simple regression that maps the stimulus representations to brain activity.

**Experimental design/evaluation:**
The pretrained models are compared to their fine-tuned models under audio-only tuned and audiovisual tuned conditions:
* *Representation similarity analysis:* This analysis evaluates the similarity between DNN embeddings across layers and EEG response under four conditions: (i) pretrained model, Audiovisual noise congruent, (i) pretrained model, Audiovisual noise incongruent, (i) fine-tuned model, Audiovisual noise congruent, (i) finetuned model, Audiovisual noise incongruent. This analysis provides a layer-wise alignment profile and tests how fine-tuning and audiovisual congruency modulate human-like information processing across layers (early auditory to higher-level language).
* *Neural encoding performance:* neural encoding models are evaluated based on similarity between DNN embeddings and predicted neural responses for acoustic (speech- and noise-envelope) and linguistic features (word onset and word surprisal). Here, the predicted neural responses are obtained from neural encoding models. This analysis provides whether the predicted responses from the neural encoding model could preserve the representational geometry of the actual neural data.

**Main findings:**
According to the authors’ interpretation, the main findings are as follows:
* Speech processing with audiovisual-congruent noise is significant for most of the EEG channels across layers in pretrained and fine-tuned models, whereas the audiovisual-incongruent condition is not.
* Feature-based DNN-brain alignment revealed that acoustic features show higher similarity in shallow layers, while linguistic features show higher similarity in deeper layers.
* For both pretrained and finetuned models, acoustic features show decreasing trend, while linguistic features show an increasing trend towards deep layers.
* The Comparison between audio-only and audiovisual tuned models reveal that the increasing trend of noise envelope shifts to layers 1-4 for audio-only and 1-5 for audiovisual, while decreasing in deeper layers.

**Strengths:**

I found this work to have the following strengths:
* *Clarity:* The stimulus conditions involving speech clips with audiovisual congruent and incongruent setups are well explained, with a clear example. The EEG dataset collection, the DNN architectures, and their feature extraction process is easy to follow, with a clear separation between pretrained and fine-tuned models. Later, the neural encoding model using mTRF estimation, and the results across three subsections are clear.
* *Originality:* The idea of using speech clips in audiovisual noise conditions while participants listen to speech, while simple, is quite novel conceptually. Since, humans process speech comprehension naturally under these natural conditions, this paper examines the DNN-brain alignment across these conditions in both pretrained and finetuned settings and provides a generalization of human-like hierarchy information processing in both systems.
* *Significance:* This work is significant in that it contributes to a better understanding of the parallels between speech language models and neuroscientific findings from speech comprehension in the human brain. It helps to understand how the layers of speech models process information and how the representation changes across layers while tuned with audio-only and audiovisual information. Overall, the new EEG dataset offers more naturalistic conditions and maintains prior findings, such as acoustic information processing in shallow layers and linguistic information in deeper layers.

**Weaknesses:**

From my perspective, the primary weaknesses of this study arise from the lack of literature works, selection of evaluation metrics, and findings from single DNN model:

* *The selection of RSA as an evaluation metric is unclear*. Prior studies report that distributional similarity measures like RSA which compare the statistical properties of neural and model representations but may not capture fine-grained functional correspondences [Marques et al., 2021]. Therefore, authors should also consider metrics like PCC and R2 score as these are commonly used in neural encoding models.
* *Limitation of model scope:* The most significant limitation is the considering of only one DNN model, which the authors acknowledge in the limitations section. However, this limitation blocks generalizability of the findings. For instance, the authors considered the AV-HuBERT model and fine-tuned it with audio-only and audiovisual settings. A strong empirical evaluation include additional models such as AST, Wav2Vec2.0 and Whisper: AST is audio spectrogram Transformer model; Wav2Vec2.0 is a self-supervised speech model; while Whisper is an encoder-decoder model in which encoder process speech while decoder performs automated speech recognition. Such comparisons would test whether these models also have similar findings as AV-HuBERT or are different in information processing of acoustic and linguistic features across layers. Further, prior studies [Miller et al. 2022] and [Vidya et al. 2022] reported that shallow layers process acoustic and phoneme-level information while deeper layers process more linguistic information. Also, authors should consider fine-tuned ASR models and evaluate their findings.
* *Fixed regularisation parameter:* The results are reported using an optimal regularisation parameter λ=0.01. In general, for a voxelwise encoding model, we learn one regularisation parameter for each voxel. However, for the mTRF model, the authors use a single regularisation parameter across 64 channels. The authors did not discuss any reasons for this selection.
* *Lack of related work:* One of the major limitations of the current study is the lack of related work where prior study explores the alignment between DNN models and brain alignment, considering the hierarchy of information processing. For eg: the introduction sentence of speech comprehension cites Hickock and Poeppel but missing popular speech comprehension work (Huth et al. 2016). Several other interpretability works, such as Oota et al. 2024, investigated the type of information present in language models that truly predict brain activity by controlling low-level features across layers of representation, and find that speech models prediction in auditory cortex is beyond low-level features while prediction in late language regions is largely due to low-level features, indicating lack of semantics. Also, Oota et. al. 2023 investigated pretrained and task-specific fine-tuned models to see how task-specific models show alignment across layers in language regions. I recommend authors to include following works:

[Huth et al. 2016] Natural speech reveals the semantic maps that tile human cerebral cortex, Nature-2016

[Oota et al. 2023] Speech Taskonomy: Which Speech Tasks are the most Predictive of fMRI Brain Activity? Interspeech-2023

[Oota et al. 2024] Speech language models lack important brain-relevant semantics, ACL-2024

[Wu et al. 2025] Distinct social-linguistic processing between humans and large audio-language models: Evidence from model-brain alignment

[Rupp et al. 2025] A hierarchy of processing complexity and timescales for natural sounds in the human auditory cortex, PNAS-2025

For a complete and detailed account of both major and minor issues, please refer to the “Questions” section.
I would like to thank the authors for the interesting naturalistic speech listening setup invested in this work. However, there are several points that I believe require further attention/work. I have divided these into major issues, which should be prioritized, and minor ones, which should be addressed for a strong version of current work.

**Questions:**

**Major Comments**

* *Impact of PCA on the RSA metric:* The authors perform PCA on stimulus features to reduce dimension from 768 to 205 and use the reduced dimension in measuring RSA between DNN embeddings and Neural Response. As I mentioned earlier in weaknesses, RSA focuses more on comparing statistical properties and may miss fine-grained functional correspondences; it remains unclear whether the findings remain similar while considering higher dimensions. I strongly encourage the authors to measure the similarity before applying PCA to verify whether the findings remain the same. In addition, since predictive metrics are common in neural encoding, I suggest authors perform similar analysis with either PCC or R2 metric, and please provide conclusions drawn from both RSA and either PCC or R2.
* *DNN model validation is missing:* The authors empirically evaluate the pretrained, and fine-tuned conditions under audio-only and audio-visual conditions and observe the alignment patterns with the brain across layers. However, the authors should perform similar analysis on several audio, audio-visual datasets to verify whether there is a task performance effect across layers. This is an important evaluation to confirm what fine tuning really adds into the model.
* *Clarity in Figure2:* Figure 2 reports only four conditions . However, the fine tuning was performed under two conditions (audio-only and audiovisual), there should be a total six topographic representational similarity maps. I recommend authors to include all six settings and clearly specify the fine-tuning settings for clarity.
* *Neural encoding with stimulus features:* The neural encoding experiment in this paper considers four interpretable features. I strongly recommend authors to consider stimulus features in the neural encoding experiment, as this approach helps if the features contain dominant acoustic information, early layers should show higher similarity; if linguistic information is dominant, deeper layers should show higher similarity with the brain. If time permits, I suggest authors perform this experiment, and referencing them in the main manuscript. This is because prior speech-based brain encoding studies that all use full feature representations from DNN used in neural encoding models.

**Minor Comments/Typos**
While addressing the following points may not be critical to the paper’s core contributions, doing so would enhance the overall quality.
* Line 48: I would recommend authors to clearly mention auditory cortex instead of primary cortex. Please follow uniformity as the primary cortex is used only here. Otherwise make it clear primary vs. non-primary cortex and use the terms throughout the paper.
* Line 148: Consider relocating the entire Training Protocol section to Appendix G to conserve space. This section does not provide method-specific insights central to the main narrative and could be referenced as needed.
* Line 52: Please clarify deeper layers are better predicted by phonetic features. Here, phonetic features are low-level phonemes or higher-level language features?
* Line 248: Explain ε(t) the noise term in the equation on line 246.
Clarify in Figure captions 2 and 3 that the numbers reported above on each Topomap denotes layer’s RSA value.
* Line 126: matrixs -> matrices
* Line 138: resulting 32 speech-in-noise -> resulting in 32 speech-in-noise
* Line 231:  1 to10 Hz ->  1 to 10 Hz
* Clarify: Line 248 says time lags of -200 to 800 ms while Line 260 says 0 to 800ms. Please clarify why these ranges are different?
* Figure 2C: Layer 12 RSA value is -0.74 while rest all are of three decimals. Please verify it once.
* Line 103: “Speech audios” is not a correct term. If we say audio, it includes non-speech and speech. Please use either speech or audio, but not both. Correct at several other places in the paper.
* Line 251: Pearson’s correlation -> Pearson correlation
* Uniformity is missing: feed forward vs. feed-forward, audiovisual vs. audio-visual, pretrained vs. pre-trained
* Line 423: LME is not defined or described anywhere in the main paper. Linear Mixed Effects only defined in Appendix.
* Line 245: EEG responses was excluded -> EEG responses were excluded

---

### Official Review · Reviewer_9sfq · 2025-10-31

**Soundness:** 2
**Presentation:** 2
**Contribution:** 2
**Rating:** 2
**Confidence:** 3

**Summary:**

This paper study a novel audio-visual EEG dataset, composed of 32 audio-visual stimuli, with 16 congruent audio-visual noise and 16 incongruent audio-visual noise. On top of these audio-visual background could be heard speech, and the 20 subjects had to answer questions on the speech or visual contents.
The paper introduces a double-step approach to model brain responses.
First, the response of each EEG electrode is predicted/denoised by fitting mTRF encoding model from basic auditory and linguistic features.
Second, the pattern of shared temporal response across stimuli of these predictions is compared to the pattern of shared temporal response across stimuli of the model (25Hz) for each electrode and model layers independently.

The main contribution of the paper is perhaps it's study of the representation of conflicting audio-visual informations, with a different encoding scores from DNN models between the congruent and incongruent case.

**Strengths:**

One of the paper main strength is to study a multimodal scenario that is in fact very common in daily life: congruent or incongruent audio-visual noise.
Studying how the brain reacts to these multi-modal inputs could help to shape our understanding of multi-modal integration in the brain.
Notably, it could help reveal how the brain segregate and combine different stream of informations, which is probably done through task-dependent selecting mechanisms.
The paper consequently present an interesting experimental paradigms, which is then dissected by modeling the coding of linguistic features.
On the results, the paper hints to several remarkable and unexpected results.
- First, audio fine-tuned model present a worse code of acoustic features than pretrained models. This question the efficacy of the audio-only model in preserving acoustic informations. In contrast the audio-visual models kept a good encoding of acoustic features while
- The paper stress large difference in the encoding of "basic features": speech enveloppe, noise enveloppe, word onset and word surprisal in fine-tuned versus pretrained models.

The paper present overall a large amount of work including:
- dataset recording
- dataset preparation (use of forced alignment through MFA)
-  exploration of novel multi-modal LLM for brain encoding.
Rigorous methods
- careful statistical estimation of the robustness of the finding, although these findings interpretation could be disputed (see weaknesses)
- cross-validations (although not fully nested)

**Weaknesses:**

Major remarks:
Despite the previous two interesting results, the author center their claim on their simulation approach. Unfortunately this approach seems more advocated by a lack of ability to record large novel dataset rather than a true need for such methodological development.
If this approach has been designed to compensate the noisiness of EEG recordings, the author should state it explicitly and motivate their use of EEG compared to other recording techniques.

More precisely, this approach cast doubts on how informative the results are:
- The author suggest that one of the main strength of their approach is to:
"enables us to generate simulated neural responses for new noise scenarios without the need to re-collect neural activity from the
human brain. Therefore, our framework can be extended to evaluate DNN-brain alignment in more realistic and diverse noisy environments."
I am not convinced that the author study true neural representations, as they mostly compare two model predictions.
The author present no evidences that studying simulated neural representations is a viable alternative to performing novel recording of brain data. To collect such evidences, the author should demonstrate that their approach is able to predict results obtained from the separate analysis of true neural recording, without these results being a direct consequence of using a model with a hierarchy of computations.

- The encoding of linguistic features in speech-processing neural network has been studied in the past (for example, Pasad et al. 2022).
Similarly, the encoding of linguistic features in the brain has also been studied in the past (for example Li et al, Nature Neuroscience 2023),  although not necessarily in the audio-visual condition and multimodal models
Relative to these published findings, the paper seems to deliver no novel insights gained from multi-modal models, but I could be partially wrong in this assessment.

- Overall, because of their simulation procedure, the paper makes conclusions on model behavior which could be potentially unrelated to the recorded neural data. Furthermore, these conclusions are focused on a hierarchical integration of acoustic to linguistic features which have been reported previously. In turn it fails to make strong conclusions on audio-visual integration which has not been extensively studied.
For example their conclusions reads "Our findings suggest that speech representations learned in transformer-based DNNs
naturally evolve toward the human-like processing patterns, providing credible evidence for the interpretability of DNNs in automatic speech recognition tasks.", which is devote of the audio-visual paradigm, of main interest here. Hierarchical speech processing through the lenses of DNNs have been considerably explored, with cleaner recording apparatus (MEG, fMRI and intracranial recording).

Minor remark:
- The paper makes claim about the representational geometry, while it compares time-courses across stimuli. The neural code for linguistic inputs is usually defined to be a population pattern, with the population responses potentially evolving in time.
It is not clear why the author decided to study time-courses across stimuli, but this could be due to the difference in EEG recording versus intracranial, fMRI or MEG recording with which the present reviewer is more used to.

**Questions:**

- It was not exactly clear how the RDMs were computed. My interpretation was that for each electrode the temporal signal of two stimuli were correlated, but for the neural-network it was unclear if this correlation was done for each unit of each layer as well.
Usually, "representations" geometry is done across unit responses at the level of the whole brain or of a ROI.
Would it be possible for the author to make this more explicit?

- Would it be possible for the author to recenter their writing and explanations on the dataset being studied rather than the model representations?
- What have they understood on audio-visual processing from this exploration?
- How do their brain recording and model representations differ in this audio-visual processing? Notably, what do the author make of the lower RDM correlation in the incongruent vs congruent condition of figure 2?

- Why was it necessary for the author to use the simulation approach?
- How can the author be certain that their simulation faithfully reproduces neural responses (no r or R2 scores reported!)

- In their TRF regression, an optimal lambda hyperparameter seems to be used. Normally these regressions use a nested-cross validation approach, with the regularization parameter potentially varying across outer cross-validation folds. Why was it impossible to follow such nested-cross validation in this scenario?

Minor question:
- Why do the author resort to PCA of the model embeddings?

---

### Official Review · Reviewer_LRek · 2025-10-31

**Soundness:** 3
**Presentation:** 2
**Contribution:** 3
**Rating:** 4
**Confidence:** 2

**Summary:**

In this paper authros explores aligment between DNN representations and human brain activity during speech processing unver audiovisual noise. Using transformer based ASR model (AV-HuBERT) and EEG recordings, the authors correlate intermediate DNN resperesentation with neural encoding responses associated with both acoustic (speech + noise envelopes) and linguistic (word onsets, word suprisal) speech features. They also investigates how the DNN-brain aligments changes before and afer fine-tuning the DNN on noisey, multimodal data.  The observe that shallow layes of the DNN align with acustic brain responses, while deep layers correspnd to lingustic responses. Fine-tuning further modulates these aligmnets, especially with multi-modal input.

**Strengths:**

(1) The work is well motivated to better understand representational parallels betwwen artifical and biological systems.

(2) The combination of neural encoding models (for generating feature-specific simulated EEG responses) and representational similarity analysis (RSA) enables the study to probe alignment across specific levels of processing in both the DNN and the brain.

(3) Comprehensive empirical evaluation is provided, including both pre-trained and fine-tuned DNNs and comparisons between audio-only and audio-visual models, strengthening the reliability of the main claims.

**Weaknesses:**

(1) No side-by-side comparison is provided between other competitive ASR architectures or alternative transformer variants.

(2) While these can be standards , but i think since only two acoustic (speech envelope, noise envelope) and two linguistic (word onset, word surprisal) features are analyzed there is a risk the results reflect the peculiarities of these specific features, rather than fundamental DNN-brain mapping

(3) The claim that fine-tuning naturally evolves DNNs toward human-like processing patterns may be somewhat overstated. For example, reductions in acoustic-feature alignment post-fine-tuning (Section 3.4, Fig. 4B, C) can also indicate loss of generalized auditory processing in favor of task-specific cues, which is not clearly discussed or  examined in the experiments.

(4) There can be potential overfitting in simulated EEG. Regularization and strategies for avoiding overfitting are not fully detailed. The more detailed ablation, or alternative validation (eg.  leave-one-subject-out) ) could increase the condidence in score.

(5) There is limited documentation on a few critical technical choices, such as the exact PCA implementation (number of components, variance explained per-layer, potential for information leakage), and whether equal-length segments are enforced after alignment/truncation. This is particularly relevant because EEG and DNN embeddings might not have matched temporal resolutions or coverage post-processing (see Section 2.5 and related).

**Questions:**

(1) Is there any justucatuin for the choice of acoustic and linguistic features ? I wonder if features (such as phonemes, syllables, or syntactic even annotaitons) can yield substantially different aligment maps ?

(2) How robust is the EEG–DNN representational alignment when applying alternative dimensionality reduction schemes beyond PCA, or when matching time windows in different ways?

(3) The fine-tuning process appears to decrease representational similarity for some acoustic features. To what extent does this tradeoff reflect task optimization, and how do the authors reconcile this with the claim of "natural" emergence of brain-like processing?

---

### Official Review · Reviewer_H2zJ · 2025-11-01

**Soundness:** 2
**Presentation:** 2
**Contribution:** 1
**Rating:** 2
**Confidence:** 3

**Summary:**

This paper investigates representational alignment between deep neural networks (DNNs) and human brain activity during speech processing under audiovisual noise. The authors employ neural encoding models (mTRF) to simulate brain responses to acoustic features (speech/noise envelope) and linguistic features (word onset/surprisal), then apply representational similarity analysis (RSA) to compare these simulated responses with embeddings from AV-HuBERT (a transformer-based ASR model). Using EEG data from 20 participants listening to speech with audiovisual noise, they demonstrate that shallow DNN layers align with acoustic processing while deeper layers align with linguistic processing. Importantly, fine-tuning on audiovisual noisy data enhances this alignment by improving noise representation in shallow layers and refining linguistic representations in deep layers.

**Strengths:**

Its primary strength lies in its methodology, which combines neural encoding models (mTRF) with Representational Similarity Analysis (RSA) to enable a precise, feature-level analysis beyond simple comparisons of raw signals to embeddings. The study is particularly well-designed due to its systematic distinction between acoustic features (envelopes) and linguistic features (word onset/surprisal), targeting established neural correlates of sensory (N1-P2) and higher-level (N400-like) processing, respectively. The research utilizes robust experimental controls, including comparisons of pre-trained versus fine-tuned models, audio-only versus audio-visual fine-tuning, and congruent versus incongruent conditions across 20 participants. Empirically, the paper convincingly demonstrates a DNN representation align with brain’s acoustic-to-linguistic processing with shallow layers (1-6) showing high similarity to acoustic features that decline with depth, while deep layers (7-11) exhibit increasing similarity to linguistic features. Finally, the modulation analysis provides a genuine insight into adaptive computational strategies by showing that audio-visual fine-tuning shifts the acoustic-linguistic transition layer and enhances noise processing, suggesting a specific mechanism for multimodal integration in noisy environments.

**Weaknesses:**

- Significant gaps in Related Works literature review
    - While the first contribution regarding methodological contribution could mislead that combination of using **neural encoding models and Representational Similarity Analysis (RSA)** is completely new, (Tuckute et al., 2023) have combined neural encoding models and RSA. While authors have cited (Tuckute et al., 2023) it wasn’t cited in a context of methodology.
    - Hierarchical correspondence between acoustic and semantic processing has been previously demonstrated in both high-temporal-resolution data, such as the findings reported by (Goldstein et al., 2025) using ECoG, and within the fMRI domain through studies of hierarchical auditory processing.
        - (Goldstein et al., 2025) develops **encoding models** to map low-level **acoustic features**, mid-level speech features, and contextual word embeddings onto brain activity during real-life conversation, demonstrating a unified framework for these feature types.
        - (Tezcan et al., 2023), which authors also cited, uses **encoding models** (specifically Temporal Response Functions, or TRFs, a type of encoding model) to investigate how **acoustic features** and **phoneme-level linguistic features** contribute to brain responses during speech comprehension.
        - While different modality(fMRI), (Heer et al., 2017, https://www.jneurosci.org/content/37/27/6539) showed **hierarchical processing** of **acoustic (spectral) & linguistic (articulation) features** through **voxelwise encoding models** on **natural speech comprehension** in **fMRI**.
- Justification for Methodology is Needed
    - This paper aims to examine the similarity between how humans adapt to noise in noisy environments and the process in encoding models. However, there's insufficient explanation as to whether the encoding model + RSA approach is absolutely necessary to examine this. Wouldn’t the traditional encoding model approach be sufficient?
- Clarifying Augmentation vs. Adaptive Alignment;
    - Clarification is needed on whether adding noise is just from a model augmentation perspective, or model actually became better at handling noise. It seems necessary to check whether similar patterns of results appear after audio augmentation with other methods. Audio flip, for example, that humans don’t naturally perceive, would be a good comparison to validate if the model actually adapted to align with human brain representation of recognition on noisy environment.
- Scope of Linguistic Features :
    - The expression 'linguistic feature' might be too risky. Linguistic feature can include far more features other than word onset, word surprisal; such as phonemes, syntactic structure.  To convincingly demonstrate that the deep layers of a speech model like AV-HuBERT capture broad linguistic features, it would be beneficial to include a comparative RDM analysis using established, high-dimensional **pretrained language model (LLM) embeddings** (e.g., from GPT or BERT). This would confirm whether the observed alignment truly extends to abstract, multimodal linguistic knowledge.

**Questions:**

- The authors should strengthen introduction to include more relevant papers and clearly denote what has been done and what novelties are added. The idea of mimicking the process by which humans adapt to noise in noisy situations by giving noise to the model and finetuning it to better predict human brain activity seems like an interesting idea, but it doesn't seem to be well highlighted.
- The importance of using RSA in combination with encoding model doesn’t seem to be clearly demonstrated. The authors should explain what limitations arise when using a general encoding model alone (i.e., merely predicting brain response from DNN intermediate representations after exposure to AV noise stimuli).
- Fine-tuning hyperparameters (learning rate, batch size, number of epochs) are not provided. It could be provided in appendix.
- Regarding experimental details, it doesn't seem to be clearly written whether what humans heard and the noise given to the model are the same. The authors should clarify this point.

---

### Note · Authors · 2025-11-20

I have read and agree with the venue's withdrawal policy on behalf of myself and my co-authors.